# Role of Some microRNA/ADAM Proteins Axes in Gastrointestinal Cancers as a Novel Biomarkers and Potential Therapeutic Targets—A Review

Agnieszka Kalita [1,2], Magdalena Sikora-Skrabaka [1,2] and Ewa Nowakowska-Zajdel [1,2,*]

1    Department of Nutrition-Related Disease Prevention, Department of Metabolic Disease Prevention, Faculty of Health Sciences in Bytom, Medical University of Silesia in Katowice, 40-055 Katowice, Poland
2    Department of Clinical Oncology, No. 4 Provincial Specialist Hospital, 41-902 Bytom, Poland
*    Correspondence: enowakowska-zajdel@sum.edu.pl

**Abstract:** Gastrointestinal (GI) cancers are some of the most common cancers in the world and their number is increasing. Their etiology and pathogenesis are still unclear. ADAM proteins are a family of transmembrane and secreted metalloproteinases that play a role in cancerogenesis, metastasis and neoangiogenesis. MicroRNAs are small single-stranded non-coding RNAs that take part in the post-transcriptional regulation of gene expression. Some ADAM proteins can be targets for microRNAs. In this review, we analyze the impact of microRNA/ADAM protein axes in GI cancers.

**Keywords:** gastrointestinal cancers; microRNA/miR; ADAM proteins; biomarkers

## 1. Introduction

Gastrointestinal (GI) cancers are some of the most common cancers in the world. GI cancers include malignancies of the GI tract (esophagus, stomach, small intestine, colon, rectum and anus) and other digestive organs (pancreas, gallbladder, liver and bile ducts) [1]. According to the International Agency for Research on Cancer's estimation, there were about 5.1 million new GI cancer cases and over 3.5 million GI cancer deaths in the world in 2020 [2]. The etiology and pathogenesis of GI cancers are multifactorial, but despite many studies worldwide, they are still unclear [3]. One of the research directions is looking into the role of ADAM proteins and microRNAs in cancer development and progression, and options for treatment as well as drug resistance.

ADAMs are a family of transmembrane and secreted metalloproteinases that play an important role in cancerogenesis, metastasis and neoangiogenesis. They have the potential to be used as prediction biomarkers or pharmaceutical targets. The role of ADAM 8, 9, 12, 17, 29 and 33 is best known in GI cancers [4,5]. ADAMs' role in carcinogenesis is associated with chronic inflammation processes. For example, the molecular targets of ADAM10 and ADAM17 in inflammation and cancer are tumor necrosis factor-alfa (TNF alfa), inteleukin-6 (IL-6), ICAM-1 and epidermal growth factor (EGF) [6–8]. ADAM28 can reactivate the activity of insulin growth factors (IGFs) in the complex of insulin growth factor binding protein-3 (IGFBP-3). IGF signaling leads to proliferation in different GI cancers [9]. Numerous studies also showed that ADAM expression is associated with poor prognosis in cancer patients [10,11]. Proteins of the ADAM family (ADAMs and ADAM with trombospodin motif—ADAMTS) play critical roles in cell–cell and cell–extracellular matrix (ECM) communication. ADAMs are membrane proteins characterized by additional EGF-like, transmembrane and cytoplasmic domains, and ADAMTSs are proteins secreted by cancer and stromal cells, characterized by an ancilliary domain containing trombospodin. ADAMTSs may contribute to modifying tumor microenvironments and are implicated in cell invasion, migration, proliferation and angiogenesis via mechanisms involved in cleaving or interacting with ECM components or regulatory factors [12]. ADAMs

and ADAMTSs have catalytic properties. Most of their substrates are membrane-bound precursors. Only 12 ADAMs, including ADAM 8, 9, 10, 12, 15, 17, 20, 21, 28, 30 and 33, have a catalytic site, and two of them (ADAM 20 and 21) have no known substrates. The substrates for the ADAMs are growth factors, chemokines, adhesion molecules and their receptors [13]. Among these ADAMs, ADAM8, 9, 10, 12, 15, 17, 19, 22, 23 and 28 have been demonstrated to play a regulatory role in the initiation, procession and metastasis of cancers [14].

In recent years, microRNAs (miRs) have been of great interest, and there are a few emerging studies on miRs and ADAMs in cancer. MiRs belong to the RNA interference family, originally discovered in 1998 by Andrew Fire and Craig Mello (2006 Nobel Prize laureates in Physiology or Medicine) [15]. About 2600 miRs have been identified in the human genome [16]. They are small single-stranded non-coding RNAs that take part in the post-transcriptional regulation of gene expression. MiRs play a fundamental role in the regulation of physiological processes such as embryogenesis as well as several human pathologies such as cancer, and auto-immune and cardiovascular diseases [17–19]. MiRs can be released into bodily fluids such as stools and blood [20,21]. They play roles in cancer biology, such as cell cycle control, metabolism, apoptosis, metastasis and angiogenesis. MiRNAs have also been introduced as promising therapeutic targets for cancer treatment [22].

Many studies revealed that some ADAMs are targets for some miRs. This review is based only on several available analyses that show correlation between miRs and ADAMs, and their role in GI cancers, and were published from 2010 to 2022. This information was searched for in databases such as Pubmed, MEDLINE and Scopus using the terms ADAM, adamalisynes, miR, cancer and malignant neoplasm. It will be interesting to learn what role miRs play in the regulation of ADAMs, how this affects tumorigenesis and what role they may play in the diagnosis and treatment, especially with regard to therapy resistance.

The purpose of this review article is to attempt to summarize the current knowledge on the role of ADAM proteins and, in particular, microRNA/ADAM protein axes in GI cancers. Understanding these mechanisms is particularly important in the search for potential biomarkers both in the diagnosis and treatment of GI cancers.

## 2. Role of ADAM Family Proteins in Gastrointestinal Tumors

The ADAM family plays a role in numerous signaling pathways associated with carcinogenesis. These include the PI3K, Notch, TGF-$\beta$, EGFR, IGF system and TNF-$\alpha$ signaling pathways. The roles of ADAM proteins relate to many of the processes involved in these pathways, such as as regulation of cellular adhesion, cellular growth, angiogenesis, inflammation and modulation of the immune system [13,14].

### 2.1. Esophageal Squamous Cell Carcinoma (ESCC) and ADAM9, 12, 17

The results of the available studies indicate the important role of ADAM9, ADAM12 and ADAM17 proteins in the pathogenesis of esophageal cancer. Zhou et al. hypothesized that ADAM12 is responsible for metastasis promotion and tumor invasion via the TM4SF3-related pathway. It has been shown to reduce the expression of selected tetraspanins by using an anti-protein antibody. ADAM12 also significantly reduces the invasiveness of esophageal squamous cancer cells [23]. Other studies confirm that ADAM17 expression is significantly higher in ESCC than in healthy tissue, and the level of expression correlates with the clinical advancement of the disease, including the presence of distant lesions [24]. ADAM9, on the other hand, modulates the pathways for vascular endothelial growth factor (VEGF), participates in angioinvasion and also correlates with cell adherence and migration [25,26].

### 2.2. Gastric Cancer (GC) and ADAM 8, 9, 10, 12, 15, 17, 33

The ADAM10 and ADAM17 proteins seem to play a particularly significant role in the pathogenesis of gastric cancer. One of the mechanisms by which these proteins contribute

to carcinogenesis is their association with the occurrence of chronic inflammation caused by *H. pylori* infection [27]. The ADAM17 protein also promotes the progression of GC through the Notch and/or Wnt-related signaling pathways [28] and the EGF-related pathway. An increase inADAM17 expression, stimulated by transforming growth factor (TGF-β), causes its transactivation and an increase in tumor cell proliferation [29]. Both ADAM10 and ADAM17 expression correlate with tumor size, metastasis and TNM stage, being a negative prognostic factor [30–32]. Other members of the ADAM family with a postulated role in the development of GC are ADAM8, ADAM9, ADAM12, ADAM15 and ADAM33 [33]. High expression of ADAM8 correlates with the size of the primary tumor, vascular invasion or the presence of metastases in lymph nodes, by affecting the level of p-ERK kinase [34]. Similarly, the level of ADAM9 expression is correlated with tumor size, local tumor invasion, the presence of lymph node metastases and TNM staging [35]. ADAM33, on the other hand, by regulating the secretion of IL-18, causes increased migration and proliferation of cancer cells [36].

*2.3. Pancreatic Cancer (PC) and ADAM 8, 9, 10, 17*

The leading role in the development of PC is attributed to the proteins ADAM10, ADAM17 ADAM8 and ADAM9. A study showed that ADAM17 is involved in the progression of pancreatic cancer from early precursor lesions to advanced invasive forms [37]. The important role of ADAM10 was confirmed by silencing the expression of this protein through genetic engineering methods and, thus, reducing the invasiveness and proliferation capacity of cancer cells [38]. ADAM8 and ADAM9, on the other hand, seem to be involved in PC tumorigenesis by being related to the response to cellular hypoxia, and these proteins are likely involved in tumor progression processes by influencing neoangiogenesis, cellular migration and further growth of cell clusters, regardless of their anchorage in the matrix. In vitro and in vivo studies have demonstrated the role of ADAM9 in the progression of pancreatic cancer by directing such processes such as angiogenesis, cell migration, matrix adhesion extracellular or tumor growth independent of anchors. These processes are likely—at least in part—related to the EGFR/MEK/ERK signaling pathway [39,40].

*2.4. Hepatocellular Cancer (HCC) and ADAM 8, 9, 10, 12, 15, 17, 28*

ADAM 8, 9, 10, 12, 17 and 28 are potential biomarkers in many cancers that are responsible for cancerogenesis, metastasis and neoangiogenesis. These roles were also confirmed in HCC [41]. In HCC, increased expression of ADAM 8 [42], ADAM9 [43], ADAM10 [44], ADAM12 [45] or ADAM17 [46] has been demonstrated. One of the recent studies on cell lines showed that high expression of ADAM8 in HCC cells contributes to cell proliferation and survival, and also induces promigratory signaling pathways independently of its proteolytic activity, thus participating in cancerogenesis and metastasis [47]. ADAM9, on the other hand, appears to mediate the migration and invasion of HCC cells induced by IL-6 [48]. The important role of ADAM10 in the pathogenesis of HCC is supported by the reduced proliferation, migration and invasion of tumor cells, both in vitro and in vivo, as a result of the downregulation of ADAM10 using an RNA silencing method [49]. One of the mechanisms by which ADAM17 promotes HCC progression is the Notch1 activation pathway [50]. Moreover, a recent analysis showed the possible role of ADAM15 as a potential HCC biomarker associated with poor prognosis [51].

*2.5. Colorectal Cancer (CRC) and ADAM 8, 9, 10, 12, 15, 17, 28*

Among the ADAM family, ADAM10 and ADAM17 have the most important role in the pathogenesis of CRC, associated with the presence of a chronic inflammatory process. ADAM10 is involved in this process via the Notch protein signaling pathway. Through dysregulation of this signaling pathway, enteritis occurs, leading to the initiation and progression of CRC [52]. ADAM17 promotes tumor growth by activating growth factors from the EGF family, as well as by affecting angiogenesis and the secretion of cytokines such as IL-6, IL-10, IL-12 or TNF alpha [53,54]. These observations were confirmed by recent

analyses showing that serum levels of ADAM10 and ADAM17 are higher in patients with CRC, and that this level correlates with the degree of differentiation of tumor cells and the presence of distant metastases [55]. The association of ADAM17 with CRC arising on the basis of inflammatory changes may also be confirmed by one recent study, where a higher concentration of ADAM17 was observed in the CRC tissue of patients with concomitant diabetes and cardiovascular diseases [56]. In addition, ADAM12 and ADAM28 proteins are involved in the pathogenesis of CRC, promoting tumorigenesis by activating pathways related to IGF. Through proteolytic activity, they are directly involved in the release of active forms of IGF [57]. Among other adamalysins that are considered to be potentially important in the development and progression of CRC, ADAM8 [58], ADAM9 [59] and ADAM15 [60] are mentioned.

A summary of the role of ADAM in GI cancers is presented in Table 1.

**Table 1.** The role of adamalysines in GI cancers. Own elaboration.

| Cancer | Adamalisine | Role of ADAM |
|---|---|---|
| Esophageal squamous cell carcinoma | ADAM9 | Neoangiogenesis, angioinvasion, cell migration [25,26] |
| | ADAM12 | Metastasis promotion and tumor invasion [23] |
| | ADAM17 | Level of expression in tissue correlates with the clinical advancement [24] |
| Gastric cancer | ADAM8 | Expression correlates with clinical advancement [34,35] |
| | ADAM9 | |
| | ADAM10 | Association with chronic inflammation caused by H. pylori infection [27], progression of GC [28,29], negative prognostic factor [30–32] |
| | ADAM17 | |
| | ADAM33 | Migration and proliferation of cancer cells [36] |
| Pancreatic cancer | ADAM8 | Neoangiogenesis, cellular migration and further growth of cell clusters [39,40] |
| | ADAM9 | |
| | ADAM10 | Invasion and proliferation of cancer cells [38] |
| | ADAM17 | Progression from early precursor lesions to advanced invasive forms [37] |
| Hepatocellular cancer | ADAM8 | Cell proliferation, migration and invasion of HCC [47–50] |
| | ADAM9 | |
| | ADAM10 | |
| | ADAM17 | |
| | ADAM15 | Potential HCC biomarker associated with poor prognosis [51] |
| Colorectal cancer | ADAM10 | Tumor growth, angiogenesis, metastasis [52–55] |
| | ADAM17 | |
| | ADAM12 | Promotion of tumorigenesis by activating pathways related to IGF [57] |
| | ADAM28 | |
| | ADAM8 | Potentially important in the development and progression [58–60] |
| | ADAM9 | |
| | ADAM15 | |

## 3. Role of Selected microRNAs in Gastrointestinal Cancers

MiRs regulate essential genes for physiological process and they are associated with cancer processes in terms of sensitivity to growth signals, apoptosis escape, angiogenesis, metastasis, invasion or inflammation and genomic instability. MiRs are found in both intracellular and extracellular regions, but what is most interesting is that some of them are circulating miRs. These are the new class of biomarkers in cancer diagnostic and treatment [16,18,22].

### 3.1. ESCC and miR-126

There are numerous experimental studies that confirm the influence of microRNAs on the tumorigenesis and metastasis of ESCC and chemo- or radiotherapy resistance. Up till now, over 150 differentially expressed microRNAs in ESCC have been identified. For example, miR-126 is identified as a tumor suppressor and a potential prognostic indicator in ESCC [61]. It is downregulated in ESCC tissues and cell lines. Haomiao Li et al. found that the expression of miR-126 is connected with clinical advancement. They suggested that miR-126 could play a role in ESCC carcinogenesis. The results of their study showed that miR-126 can function as a tumor suppressor via the regulation of insulin receptor substrate 1 (IRS1) and GOLPH3 [62]. The PI3KR2 gene, which regulates the PI3K/AKT signaling pathway, was found to be a potential target for miR-126. This axis can inhibit ESCC invasion [63]. MIR-126 was also found to be a regulator of the PTPN9 protein, whose expression is connected with inhibiting cancer development [64]. In contrast, miR-126 is suggested to suppress apoptosis and autophagy by targeting STAT3. This study shows that miR-126 can promote ESCC development by inhibiting cell death [65]. The level of plasma miR-126 was not associated with clinicopathological features and clinical outcomes in patients with ESCC [66].

### 3.2. GC and miR-126, miR-320

MiR-126 is also downregulated in GC cells and tissues. A number of studies showed that miR-126 inhibits the development and invasion of GC by targeting, among others, the IGF-1R [67], GOLPH3 [68], VEGF-A [69], Crk protein [70,71], SRPK1 [72], CRKL [73] or LAT-1 genes [74]. In contrast, miR-126 may promote the proliferation and invasion of GC, by downregulation of CADM1 [75].

Runhua F. et al. found that serum miR-126 level in patients with GC is downregulated too, and this is connected with aggressive progression and poor prognosis. Additionally, in patients with locally advanced but lymph node-negative GC, downregulation of miR-126 is an unfavorable prognosis factor. Furthermore, miR-126 can be associated with OS and DFS in patients with GC. [76–78].

Jin Y. et al. discovered that the expression of long noncoding RNAs (lncRNAs) HOX antisense intergenic RNA (HOTAIR) in gastric cells and tissues is associated with cancer progression and chemoresistance. They found that HOTAIR promotes cisplatin resistance in GC cells by activating the PI3K/AKT/MRP1 pathway after the inhibition of miR-126 expression. Ping W. et al. also found that the downregulation of miR-126 can enhance vincristine and adriamycin resistance in GC cell lines. These results suggest that miR-126 can increase the sensitivity of GC cells to chemotherapy [79,80]. Varkalaite et al. proved that miR-129 is significantly downregulated in GC cells compared to a healthy control group; therefore, it is suspected to be a potential early diagnostic biomarker. Moreover, it can be associated with the prognosis of GC patients and correlates with an increase in the number of metastatic lymph nodes. MiR-129 is downregulated in GC cells, tissues and plasma, and its level depends on malignancy, whereas oncogene HOXC10 is upregulated. Yu et al. confirmed that HOXC10 is a direct target for miR-129 and this axis regulates the apatinib resistance of GC cells [81–84]. MiR-129 inhibits GC progression and proliferation by targeting, among others, the high-mobility group protein B1 (HMGB1) [85], COL1A1 [86], SAE1 [87], WW domain-containing E3 ubiquitin protein ligase 1 (WWP1) [88], SPOCK1 [89], HOXC10/Cyclin D1 [90], BDKRB2 [91] and IL-8 [92,93].

MiR-320 is downregulated in GC cell lines and regulates, among others, the KLF5/HIF-1$\alpha$ signaling pathway, which is responsible for cell migration, invasion and epithelial–mesenchymal transition (EMT). Zhou et al. found that this process can be inhibited by targeting the miR-320/KLF5/HIF-1$\alpha$ pathway. It can also suppress GC development by targeting FoxM1 [94,95]. MiR-320 can be a target for small nucleolar RNA host gene 12 (SNHG12), which can modulate CRKL expression. SNHG12 is suspected to act as a promoter of GC progression by regulating the ERK and AKT pathways [96,97].

Plasma miR-320 level is also decreased in GC patients and correlates with clinical severity indicators such as TNM stage. Moreover, miR-320 can distinguish patients with GC from healthy controls, which makes it a potential diagnostic and prognostic biomarker [98].

Numerous studies revealed that miR-338, which is downregulated in GC patients, may act as a tumor, metastasis and EMT suppressor or enhance chemosensitivity, for example, by targeting PTP1B [99], P-REX2a through a PTEN/AKT axis [100], SOX5 and blocking the Wnt/β-catenin signaling pathway [101], ZEB2 and the MACC1/Met/Akt pathway [102], ACBP-3 [103], SSX2IP [104] and NRP1 [105]. Moreover, it can also take part in cisplatin resistance by targeting ZEB2 and 5-FU resistance by targeting the LDHA-glycolysis pathway. Liu et al. demonstrated that some miRNAs, among them miR-338, might be biomarkers for the sensitivity to radiochemotherapy in patients with locally advanced GC [106–108].

### 3.3. PC and miR-126, miR-328

MiR-126 is downregulated in PC, like in other GI cancer tissues, which is related to the carcinogenesis of PC, and its low level in tissue predicts poor overall survival. Shuoling Chen et al. found that the COL12A1 and COL11A1 genes, which are connected with metastasis by acting on the ECM–receptor interaction pathway, are targets for miR-126. The KRAS signaling pathway, which promotes the progression of PC, is directly regulated by downregulated miR-126. Additionally, the TGF-β signaling pathway may be regulated by downregulated miR-126 [109–112]. In their study, Khakinezhad Tehrani F et al. found that miR-126 may enhance silybin encapsulated in polymersome treatment in PANC-1 cell culture [113]. MiR-328 is suspected to be a novel biomarker in PC because its level is significantly associated with OS. Based on the STRING online database and Cytoscape, Liang L. et al. predicted possible target genes that could be regulated by miR-328. Among these targets, they found well-known genes such as EGFR, MAPK1, ESR1, SMAD4 and AR, which are connected with cancerogenesis. However, there is a need for future research [114].

### 3.4. HCC and miR-145, miR-224, miR-3163

Numerous miRNAs are recognized as novel biomarkers in HCC development, invasion, diagnostics or drug resistance [115]. MiR-122 is described as a tumor suppressor in HCC patients, and its downregulated level is related to poor differentiation of HCC cells, advanced TNM stage and poor prognosis [116]. Circulating miR-122 was found to be significantly increased in early-stage HCC in comparison with healthy controls or patients with other liver diseases, such as cirrhosis or liver metastasis. The results improved the correlation with AFP level, which makes miR-122 a potential diagnostic biomarker [117]. Xu et al. proved that the overexpression of miR-122 may inhibit HCC cells' proliferation and enhance their radiosensitivity through regulation of cyclin G1. [118] MiR-122 can also enhance the sensitivity of HCC to oxaliplatin by targeting the Wnt/β-catenin pathway [119]. MiR-126 is also found in HCC cells and tissues; it is downregulated, plays an important role in HCC development, invasion and distant metastasis and is related to poor OS and DFS [120]. MiR-126 is mainly described as a tumorigenesis and metastasis suppressor by targeting, among others, LRP6 and PIK3R2 [121], Sox2 [122], EGFR [123] and EGFL7 [124]. MiR-126 is suggested to act in sorafenib resistance via targeting SPRED1 [125].

MiR-145 is downregulated in HCC tissues and cell lines. Its low level is correlated with metastasis capabilities such as cell proliferation, migration or invasion, and it is an independent poor prognostic factor [126]. MiR-145 may inhibit metastasis by targeting, among others, ARF6 [127], the ROCK1/NF-κB signaling pathway [128], IRS1 [129], FSCN1 [130] or Kruppel-like factor 5 (KLF5) [131]. Additionally, in an in vitro study, it was found that the hsa_circ_0001955/miR-145-5p/NRAS axis can act as an oncogenic pathway and be related to HCC progression. Targeting these pathways may be potentially therapeutic [132].

Additionally, the mTOR/miR-145/GOLM1 signaling pathway may be targeted for HCC treatment [133]. A low level of miR-145 is related not only to HCC progression but also to sorafenib resistance. It is also suspected to play a role in radioresistance and chemoresistance to 5-FU of HCC by targeting TLR4 and to be associated with the effects of ribavirin on HCC [134–137].

MiR-224 is overexpressed in the tissues and serum of HCC patients, and it is suspected to play a role in HCC development and invasion. An F. et al. found that IL-6/STAT3/miR-224/SMAD4 can be the new signaling pathway in HCC progression [138]. MiR-224 can also promote HCC invasion by activating the AKT signaling pathway, targeting the Homeobox D 10 gene [139]. Moreover, it is expected that the plasma level of miR-224 can be used as a biomarker for the early development of HCC. Numerous studies showed that its level was related to serum AFP level and liver damage. A high level of miR-224 was correlated with poor prognosis [140–142].

MiR-3163 is downregulated in HCC cell lines and tissues, and its low level is related to greater clinical advancement. Shi et al. confirmed that VEGF A is a target gene for miR-3163, and its overexpression is associated with invasion and angiogenesis [143].

*3.5. CRC and miR-143, miR-145, miR-195-5p, miR-17,miR-19, miR-20, miR-9, miR-497-5p, miR-217, miR-182, miR-135b, miR-125a-3p, miR-198*

CRC is the third most commonly diagnosed cancer in the United States every year, and it is the most common cancer of all GI cancers worldwide. Despite diagnostic tools such as colonoscopy, about 25% of patients are diagnosed at an advanced stage of the disease [144]. As of today, we know some molecular abnormalities, such as RAS or BRAF mutations, that can be prognostic and predictive biomarkers [145].

The role of miRNA in regulating CRC signaling is fundamental to understanding the processes of progression and cancer angiogenesis. MiRNAs also play an important role as diagnostic biomarkers in CRC. It is important to focus on the signaling pathways leading to CRC and the role of miRNAs in their regulation. Sporadic CRC can develop through two molecular pathways. The suppressor gene accumulation pathway is characterized by chromosomal instability (CIN) and the other pathway results from DNA repair gene dysfunction, on the basis of microsatellite instability (MSI). The study of Sluttetery et al. shows that significant differences in miRNA expression are associated with these pathways [146]. For the past 20 years, research has been ongoing on the relationship between the expression of various miRNAs and the progression in CRC. The most important miRNAs described in numerous studies are miRNA-143, 145, 195-5p, 17, 19, 20, 9, 497-5p, 217, 182, 135b, 125a-3p and 198 [15,147]. For example, miRNA-145 inhibits, among others, IRS1, proto-oncogenes such as c-Myc and Yamaguchi sarcoma viral oncogene homolog1 (Yes-1) and signal transducers and activators of transcription 1 (STAT1) [148–150]. MiR-143 expression is likely to be negatively correlated with CRC metastasis [151]. Additionally, the level of miRNA is measured not only in tumor tissues or cell lines, but also in blood serum. M. Radanova et al. found that the level of miR-618 circulating in the blood is significantly increased in patients with advanced CRC in comparison with healthy controls. In contrast, the expression of miR-618 in colon cancer tissue was significantly decreased. Low expression of circulating miR-618 in patients suffering from metastatic CRC is connected with shorter overall survival. The results of this study indicate that the level of miR-618 in the blood can be tested as a prognostic biomarker in patients with metastatic CRC [152].

Examples of some target genes/pathways for selected miRs in GI cancers are presented in Table 2.

**Table 2.** The role of selected miRs and their targets in GI cancers. Own elaboration.

| miRNA | Cancer | Target Genes/Pathways | Functions |
|---|---|---|---|
| miR-126 | ESCC | IRS-1, GOLPH3, PTPN9, PI3K/AKT signaling pathway | Inhibition of cell proliferation, migration and invasion [62–64] |
| | | STAT3 | Inhibition of apoptosis and autophagy [65] |
| | GC | IGF-1R, GOLPH3, VEGF-A, Crk protein, SRPK1, CRKL, LATS-1 genes | Inhibition of cell proliferation, migration and invasion [67–74] |
| | | CADM1 | Promotion of cancer development and invasion [75] |
| | PC | COL12A1 and COL11A1 genes, KRAS and TGF-β signaling pathway | Regulation of metastasis and cancer development [109–112] |
| | HCC | LRP6 and PIK3R2, Sox2, EGFR and EGFL7 SPRED1 | Inhibition of tumorigenesis and metastasis [121–124] Acting on sorafenib resistance [125] |
| miR-129 | GC | HOXC10 | Regulation of apatinib resistance [84] |
| | | HMGB1, COL1A1, SAE1, WWP1, SPOCK1, HOXC10/Cyclin D1, BDKRB2 and IL-8 | Inhibition of GC progression and proliferation [85–93] |
| miR-320 | GC | KLF5/HIF-1α signaling pathway, FoxM1 | Inhibition of cell migration, invasion and epithelial–mesenchymal transition (EMT) [94,95] |
| miR-338 | GC | PTP1B, P-REX2a through the PTEN/AKT axis, SOX5 and blocking Wnt/β-catenin signaling pathway, ZEB2 and MACC1/Met/Akt pathway, ACBP-3, SSX2IP and NRP1 | Inhibition of tumor growth, metastasis and EMT [99–105] |
| | | ZEB2 LDHA-glycolysis pathway | Regulation of cisplatin resistance [106], 5-FU resistance [107] |
| miR-328 | PC | EGFR, MAPK1, ESR1, SMAD4 and AR | Regulation of cancer development and invasion [114] |
| miR-122 | HCC | Cyclin G1 | Inhibition of HCC cells' proliferation and enhancing their radiosensitivity [118] |
| | | Wnt/β-catenin pathway | Enhancing sensitivity to oxaliplatin [119] |
| miR-145 | CRC HCC | IRS-1, c-Myc, Yes-1 and 1 STAT1 | Inhibition of cancer development [148–150] |
| | | ARF6, the ROCK1/NF-κB signaling pathway, IRS1, FSCN1, KLF5 | Inhibition of metastasis [127–131] |
| | | TLR4 | Acting on radio- and chemoresistance to 5-FU [135,136] |
| miR-224 | HCC | AKT signaling pathway through Homeobox D 10 gene | Promotion of cancer development and invasion [139] |
| miR-3163 | HCC | VEGF A | Promotion of invasion and angiogenesis [143] |

## 4. Role of miRNA/ADAM Protein Axes in Gastrointestinal Cancers

### 4.1. ESCC and miR-126/ADAM9

Liu et al. found that miR–126 overexpression in tumor tissue suppressed ESCC development and progression by inhibiting the activation of the ADAM9–EGFR–AKT pathway. This study confirmed that ADAM9 functions as a direct target of miR-126 and contributed to miR-126 repressing cell migration in ESCC [153]. The most important goal of this research was to prove that the regulation of some ADAM/miR axes may have therapeutic potential in the treatment of ESCC.

### 4.2. GC and miR-126, miR-129-5p/ADAM9, miR-338-3p/ADAM17, miR-320a/ADAM10

Based on scientific reports, the correlation between miRNA and ADAMs seems to be a strong factor in GC development. In a study carried out by Wang et al., ADAM9 was overexpressed in GC tissues, and its high levels were significantly correlated with more advanced GC clinicopathological features, such as local advancement and metastasis, described in the TNM system. MiR-126 was downregulated in GC cells. The results of this study suggest that ADAM9 is one of the targets regulated by miR-126 in GC cells and in this process, miR-126 performs its potential tumor suppressive function in GC [35,154].

Liu et al. found that miR-129-5p functions as a tumor suppressor in GC progression, also via targeting ADAM9. Their study showed that the levels of miR -129-5p are lower in GC tissues and cell lines than in cancer-free controls, which can be associated with poor prognosis of GC patients [155]. In their study, Chen et al. demonstrated that miR-338-3p level is significantly decreased both in GC tissues and cell lines and its progression is partially inhibited via the downregulation of ADAM17. This metalloproteinase regulates the release of TNF-α and ligands of EGFR from cells. ADAM17 was identified as a direct target of miR-338-3p [156]. The miR-338–3p/ADAM17 axis is also regulated by circular RNA circ_0051620, the overexpression of which is associated with GC metastasis and poor prognosis [157]. Ge et al. reported that the overexpression of ADAM10 and decreased level of miR-320a in GC cell lines. Their investigation indicates that 3'-UTR of ADAM10 is the target of miR-320a. The aim of their work was also to study the influence of the miR-320a/ADAM10 axis regulation on the sensitivity of GC cells to cisplatin. They proved that miR-320a overexpression in GC cells increases their sensitivity to cisplatin. Their findings suggest that ADAM10 is a functional target of miR-320a in GC development and chemoresistance [158].

These presented studies revealed that increased miR-126, miR-129-5p, miR-320a and miR-338-3p suppressed GC cell proliferation via the regulation of ADAM-dependent pathways, which indicates that they may be a potential target for GC therapeutic treatment.

### 4.3. PC and miR-126/ADAM9, miR-328/ADAM8

PC is one of the deadliest cancers in the world. The etiopathogenesis is still unclear despite many investigations worldwide. Currently, surgical resection is the only option for a cure, but over 75% of PC cases are unresectable at the time of diagnosis [159]. This is why we very much need diagnostic and treatment options that may improve overall survival in patients suffering from PC.

In PC cell lines, MiR-126 functions as a tumor suppressor via the regulation of ADAM9, which was confirmed by Hamada et al. The miR-126/ADAM9 axis plays a role in cellular migration, angiogenesis and invasion of PC cells, which is crucial in metastasizing [160]. Yu et al. found that propofol suppressed the development of PC cell lines through the downregulation of ADAM8 via the overexpression of miR-328. In this study, the ADAM8/miR-328 axis was identified as a novel pathway of PC progress. The overexpression of ADAM8 was found in some GI cancers such as pancreatic, colon or gastric cancers and was a negative prognostic factor. This study suggests that propofol may be one of the PC treatment options in the future [161]. The miR/ADAM axis is a promising direction in research on PC development, treatment and diagnosis. The presented studies seem to indicate new potential options for treatment.

### 4.4. HCC and miR-122, 145, 3136/ADAM17, miR-203, 1274-a/ADAM9

HCC is the primary liver cancer and is closely related to chronic viral hepatitis caused by the hepatitis B or C virus. It is also associated with excessive alcohol use and other chronic liver diseases that lead to cirrhosis. Despite commonly known risk factors, the prognosis is poor because of late diagnosis of HCC worldwide [162]. ADAM17 is overexpressed in many cancers. Investigators identified ADAM17 as a target for miR-122, 145 and 3163. Wei-Chih Tsai et al. found that miR-122 is suspected to be a tumor suppressor because its level is downregulated in HCC tissues and cell lines, just as in GC cells. The restoration of miR-122 via the downregulation of ADAM17 caused a reduction in tumorigenesis, angiogenesis and invasion [163]. Yuwu Liu et al. found that miR-145 and 224 expression is higher in HCC tissue than in normal liver cells. They investigated whether or not there is a connection with the overexpression of ADAM17. However, the correlation was not confirmed. ADAM17 is not the target for miR-145 and 224 [164]. In contrast to their previous study, they found that miR-145 is downregulated in HCC and is considered as a tumor suppressor activated via ADAM17 [165]. This is why miR-122 and 145 can be potentially curative, although the role of miR-145 is still unclear. Further research is necessary. Additionally, it has been well established that miR-122 reduces chemoresistance [116].

ADAM9, which is connected to tumor cell proliferation, invasion and inhibition of apoptosis, is also upregulated in HCC cells. MiR-203 can be described as a tumor suppressor through downregulating ADAM9 [166]. The miR-126/ADAM9 axis is described in HCC and is connected with cancer progression. Moreover, Le-yang Xiang et al. found that miR-126 is downregulated in HBV-related HCC patients without pre-operational treatment, which causes tumor development and progression by targeting ADAM9 [167].

Bin Yang et al. found that the MiR-3163/ADAM17 axis regulates the Notch pathway, which takes part in the sensitivity of HCC cells to targeted molecular treatment, such as sorafenib, and they suggest that miR-3163 can enhance this sensitivity [168]. MiR-1274-a is upregulated in HepG2 cells after sorafenib treatment, while ADAM9 is downregulated and its expression can be also suppressed by miR-1274-a [169]. These studies show other therapeutic potential for miRs.

### 4.5. CRC and miR-30c/ADAM19, miR-198/ADAM28, miR-20b/ADAM9

Both ADAMs and miRs have been intensively researched in the context of the pathogenesis of CRC, but the correlation between them is still poorly known. Wang J et al. found that the expression of miR-552 and miR-592 is upregulated in tumor tissues and cell lines and correlates with the advancement of the disease described by TNM. They also proved that miR-552 additionally promotes CRC metastasis by targeting ADAM 28 [170]. ADAM19 is upregulated in renal cell carcinoma and primary brain tumors and plays a role in the pathogenesis of various diseases, such as chronic obstructive pulmonary disease [171,172]. MiR-30c was identified as a tumor growth, migration and invasion suppressor targeting ADAM19, which is also overexpressed in CRC tissues. However, this study shows that ADAM19 is not the only target for miR-30c because the cancer development rate was greater than the reduction in ADAM19. MiR-30c is suggested to be one of the therapeutic possibilities in CRC therapy [173]. The JAK/STAT pathway promotes the progression of cancers through, for example, cell proliferation, invasion and migration. In a study carried out by L.-X. Li et al., it was shown that miR-198 inhibits CRC progression by regulating the ADAM28/JAK-STAT signaling pathway. MiR-198 is detected as a tumor suppressor [147]. MiR-20b is downregulated in colon tumor tissue versus normal tissue. In their study, Qiang Fu et al. studied the mechanism of chemoresistance to 5-FU in colon cancer. 5-FU is the standard component of chemotherapy in CRC and improves the overall survival of patients with CRC. They found that miR-20b can reduce 5-FU resistance by inhibiting the ADAM9/EGFR pathway. This result indicates that miR-20 can be a promising direction of future studies and therapy [174,175]. These studies show that some miRs can be used as prognostic or predictive factors and, in correlation with adamalisines, can play a role in CRC treatment.

## 5. Conclusions

This review shows some correlations between ADAMs and miRNAs and their roles in GI cancers. Currently, the search for targets for miRs and their functions in oncology is very popular. The presented studies reveal that some ADAM/miR axes can not only regulate cancerogenesis and metastasis but also may have therapeutic potential in GI cancer treatment or be biomarkers. All of the presented studies were carried out on cell lines or tissues. The most-known is the ADAM9/miR-126 axis, which was described in ESCC, GC, HCC and PC, and can function as a tumor- and-metastasis suppressing pathway. Moreover, some of these axes work on the EGFR, VEGF or JAK-STAT pathways, which are confirmed to be involved in tumorigenesis, malignancy and chemosensitivity. This is important not only for the biological function of miRNAs in cancer processes, but also for their potential role as biomarkers with the possibility of assessing prognosis. Such studies require not only the development of methodologies for classical biomarkers, but the establishment of algorithms for management. On the other hand, miRNAs can be used as a therapeutic target with sensitization to the chemotherapy used. MiRNA-374b in HCC [176] and miRNA-1271 in CRC play a role in the cell signaling pathway [177], while miRNA-122 in GC is important in the DNA damage repair-related genes [178].

Currently, there are only a few ADAM/miR axes known to take part in GI cancer pathology (Table 3). Unfortunately, these correlations in GI cancer locations such as the bile ducts, gall bladder and small intestine are still unknown. Therefore, continuing research on the role of ADAMs, miRNA and their correlations is fully justified. This can help to discover other cancerogenesis pathways, predictive and prognostic factors or new options for treatment. There are only a few active clinical trials with miRs, not only in oncology but also in other medical specialties such as cardiology. It seems that the clinical significance of miR/ADAM axes can be intensively studied in the near future.

**Table 3.** The role of ADAM/MiR axes in gastrointestinal cancers. Own elaboration.

| Cancer | Adamalysine | miRNA | Role of ADAM/miR Axis |
|---|---|---|---|
| Esophageal squamous cell carcinoma<br>Gastric cancer<br>Pancreatic cancer<br>Hepatocellular cancer | ADAM9 | miR–126 | Supressing tumor development and progression by inhibiting cell migration, invasion and angiogenesis [35,153,154,160,167] |
| Gastric cancer | ADAM9 | miR-129-5p | Supressing cell migration and invasion by targeting interleukin-8 [155] |
| | ADAM17 | miR-338-3p | Supressing proliferation, migration and invasion of cancer cells [157] |
| | ADAM10 | miR-320a | Progression of cancer and resistance to cisplatin [158] |
| Pancreatic cancer | ADAM8 | miR-328 | Supressing development of cancer through propofol [161] |
| Hepatocellular cancer | ADAM17 | miR-122 | Reduction in tumorgenesis, angiogenesis, invasion and chemoresistance [163] |
| | | miR-145 | Potentially supressing development of cancer [164,165] |
| | | miR-3163 | Enhancing sensitivity of cancer cells to sorafenib [168] |
| | ADAM9 | miR-203 | Supressing proliferation, migration and invasion of cancer cells [166] |
| | | miR-1274-a | Sensitivity of cancer cells to sorafenib [169] |

**Table 3.** *Cont*.

| Cancer | Adamalysine | miRNA | Role of ADAM/miR Axis |
|---|---|---|---|
| Colorectal cancer | ADAM28 | miR-552 | Promoting metastasis [170] |
| | | miR-198 | Inhibiting cancer progression by regulating JAK-STAT signaling pathway [147] |
| | ADAM19 | miR-30c | Supressing tumor growth, migration and invasion [173] |
| | ADAM9 | miR-20b | Reducing 5-FU resistance by inhibiting EGFR pathway [174,175] |

**Author Contributions:** Conception and design of the article: A.K. and E.N.-Z.; Literature search: A.K., E.N.-Z. and M.S.-S.; Interpretation of the relevant literature: A.K., E.N.-Z. and M.S.-S.; Article draft: A.K. and E.N.-Z.; Revision of the article for intellectual content: E.N.-Z. All authors have read and agreed to the published version of the manuscript.

**Funding:** This research was funded by the Medical University of Silesia in Katowice (Poland), grant number PCN-1-101/K/2/Z.

**Institutional Review Board Statement:** Not applicable.

**Informed Consent Statement:** Not applicable.

**Data Availability Statement:** Not applicable.

**Conflicts of Interest:** The authors declare no conflict of interest.

## Abbreviations

| | |
|---|---|
| ACBP-3 | Anticancer bioactive peptide-3 |
| ADAM | A disintegrin and metalloproteinase |
| ADAMTS | ADAM with trombospodin motif |
| Akt | Protein kinase B |
| AR | Androgen receptor |
| ARF6 | ADP Ribosylation Factor 6 |
| BDKRB2 | Bradykinin Receptor B2 |
| CADM1 | Cell Adhesion Molecule 1 |
| CEA | Carcinoembryonic antigen |
| COL1A1 | Collagen Type XI Alpha 1 Chain |
| COL11A1 | Collagen Type XI Alpha 1 Chain |
| COL12A1 | Collagen Type XI Alpha 1 Chain |
| CRC | Colorectal cancer |
| CRK | CRK proto-oncogene |
| CRKL | CRK-like proto-oncogene |
| ECM | Cell–extracellular matrix |
| EGF | Epidermal growth factor |
| EGFR | Epidermal growth factor receptor |
| EMT | Epithelial–mesenchymal transition |
| ERK | Extracellular signal-related kinase |
| ESCC | Esophageal squamous cell carcinoma |
| ESR1 | Estrogen Receptor 1 |
| FOXM1 | Forkhead Box M1 |
| FSCN1 | Fascin Actin-Bundling Protein 1 |
| GC | Gastric cancer |

| | |
|---|---|
| GI cancers | Gastrointestinal cancers |
| GOLPH3 | Golgi Phosphoprotein 3 |
| H. pylori | Helicobacter pylori bacteria |
| HCC | Hepatocellular cancer |
| HIF-1$\alpha$ | Hypoxia-inducible factor |
| HMGB1 | High-mobility group protein B1 |
| HOTAIR | HOX antisense intergenic RNA |
| HOXC10 | Homeobox C10 |
| ICAM-1 | Intercellular Adhesion Molecule 1 |
| IGFs | Insulin growth factors |
| IGFBP-3 | Insulin growth factor binding protein-3 |
| IL | Interleukin |
| IRS-1 | Insulin Receptor Substrate 1 |
| JAK-STAT pathway | Janus kinase—signal transducer and activator of transcription protein pathway |
| KRAS | Kirsten rat sarcoma virus is a gene that provides instructions for making a protein called K-Ras |
| KLF5 | Krüppel-like factor 5 |
| LATS-1 | Large Tumor Suppressor Kinase 1 |
| LRP6 | LDL Receptor-Related Protein 6 |
| MACC1 | MET Transcriptional Regulator MACC1 |
| MAPK1 | Mitogen-Activated Protein Kinase 1 |
| MET | MET Proto-Oncogene, Receptor Tyrosine Kinase |
| mRNA | Messenger RNA |
| miRNA = microRNA = miR | Small non-coding RNA molecule |
| NF-$\kappa$B | Nuclear factor kappa light-chain-enhancer of activated B cells |
| NRP1 | Neuropilin 1 |
| OS | Overall survival |
| P-REX2a | Phosphatidylinositol 3,4,5-trisphosphate RAC exchanger 2a |
| PC | Pancreatic cancer |
| PFS | Progression-free survival |
| PI3KR2 | Phosphoinositide-3-Kinase Regulatory Subunit 2 is a Protein Coding gene |
| PTEN | Phosphatase and Tensin Homolog |
| PTPN9 | Protein Tyrosine Phosphatase Non-Receptor Type 9 |
| ROCK1 | Rho-Associated Coiled Coil Containing Protein Kinase 1 |
| SAE1 | SUMO1 Activating Enzyme Subunit 1 |
| SMAD4 | SMAD Family Member 4 |
| Sox2 | SRY-Box Transcription Factor 2 |
| SOX5 | SRY-box transcription factor 5 |
| SPOCK1 | SPARC (Osteonectin), Cwcv And Kazal-Like Domains Proteoglycan1 |
| SPRED1 | Sprouty-Related EVH1 Domain Containing 1 |
| SRPK1 | Serine-arginine protein kinase 1 |
| SSX2IP | Synovial Sarcoma X breakpoint 2 Interacting Protein |
| STAT3 | Signal transducer and activator of transcription 3 |
| TGF-$\beta$ | Transforming growth factor beta) |
| TLR4 | Toll-like receptor 4 |
| TM4SF3 | Transmembrane-4-l-six-family-3 |
| TNF-$\alpha$ | Tumor necrosis factor alfa |
| TNM | Tumor, nodules, metastases scale |
| VEGF | Vascular endothelial growth factor |
| WWP1 | WW domain-containing E3 ubiquitin protein ligase 1 |
| Yes-1 | Yamaguchi sarcoma viral oncogene homolog1 |
| ZEB2 | Zinc Finger E-Box Binding Homeobox 2 |

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
