# Peer review of "Role of Some microRNA/ADAM Proteins Axes in Gastrointestinal Cancers as a Novel Biomarkers and Potential Therapeutic Targets—A Review"

_cimb, doi:10.3390/cimb45040191_

Round 1

Reviewer 1 Report

Comments and Suggestions for Authors

The paper is about a very interesting novel with great therapeutic implications for the future topic. It shows a lot of previous studies and passion for the topic. Congratulation on the manner of data exposing and the manner of manuscript conceptualization. 

Some regards:

- are necessary some English changes (eg: lines 31, 40)  

- the references need to be edited and carefully checked (eg: ref nr 38 is missing authors)

Author Response

Response to Reviewer 1 Comments

Thank you very much for your review, warm words and the time spent on reviewing our manuscript. Below we present the answer to the review.

Point 1. - are necessary some English changes (eg: lines 31, 40)  

Our manuscript has been proofread in English.

Point 2. - the references need to be edited and carefully checked (eg: ref nr 38 is missing authors)

Ref.nr 38 has been completed.

Reviewer 2 Report

Comments and Suggestions for Authors

Thank you for sending me the research article paper “Role of some microRNA /ADAM protein axes in gastrointestinal cancers as novel biomarkers and potential therapeutic targets– review” for review in the Current Issues in Molecular Biology. In the article Kalita et al., analyzed the role of miRNA and ADAM in the development of gastrointestinal cancers and act as therapeutic biomarkers. There are important points that should be discussed and improved.

1.      Heading: A role of ADAM family proteins in gastrointestinal tumors - a brief review: Author should explain the more details of ADAM protein role in each GIT cancers. It would be good to explain the mechanism of ADAMS protein in the development of GIT cancers.

2.      Author should divide the heading into more sub-heading according to their mechanism,miRNA and ADAMS in each cancer type heading. It would be easy and understandable for the readers. In the current stage review looks similar heading and just based on the cancers type.

3.      Author should provide a graphical presentation of the role of ADAMs with miRNA in the development of each GIT cancers (1 or 2 figures).

4.      Author should write a heading with a little more description rather than only cancer names.

Author Response

Response to Reviewer 2 Comments

Thank you very much for any comments on the manuscript and the time spent on reviewing our article. Below we present the answer to the review.

Point 1.

Heading: A role of ADAM family proteins in gastrointestinal tumors - a brief review: Author should explain the more details of ADAM protein role in each GIT cancers. It would be good to explain the mechanism of ADAMS protein in the development of GIT cancers.

The authors added some of the mechanism of action of ADAMs in tumors both in the subsection and when describing the listed cancers. However, a detailed description and elaboration of all mechanisms involved in the pathogenesis of GI cancers is beyond the scope of this manuscript and is a subject for a separate review. In the current study, the authors wanted to focus on the role of microRNA and role of miRNAs/ADAM proteins axes in GI cancers.

Point 2.

Author should divide the heading into more sub-heading according to their mechanism,miRNA and ADAMS in each cancer type heading. It would be easy and understandable for the readers. In the current stage review looks similar heading and just based on the cancers type.

The authors respected the reviewer's comment and added ADAM proteins and miRs which are relevant to specific cancers in the headings and described in the text.

Point 3.

Author should provide a graphical presentation of the role of ADAMs with miRNA in the development of each GIT cancers (1 or 2 figures).

The authors preseted the role of ADAMs and miRs and axis of ADAM/miR in table 1, 2 and 3

Point 4.

Author should write a heading with a little more description rather than only cancer names.

The authors respected the reviewer’s comments as point 2.

Reviewer 3 Report

Comments and Suggestions for Authors

Thank you for your valuable manuscript. You tried to define microRNAs related to ADAM protein as novel biomarkers in different types of GI. However, some issues need to be clarified. here, I suggest some considerations that could improve the manuscript. 

- Although the last paragraph of the introduction part should introduce the aim of your review article and explain the importance of this study, I could not find the proper description there. Please talk about why this topic was chosen as a narrative review and describe what you discussed. 

- In my opinion, the order of the manuscript is confusing and repeated. I advise you to change this order according to the following: first, define each type of GI, then define the role of ADAM in GIs, after that, it is better to define the role of miRs in GIs, and finally talk about ADAM/related miRs in GIs.

- The conclusion part does not include references or any table. I advise you to add a new session and separate a new conclusion part.

- The references were manually entered and their format is not similar. Please complete reference 9. I also kindly suggest you add references via Endnote/Mendeley that could prevent multiple errors.

- This article includes 3 tables but lacks proper descriptions and the definition of abbreviations. In the conclusion part, you mentioned table 2 instead of table 3. Please correct it.

Author Response

Response to Reviewer 3 Comments

First of all, thank you very much for any comments on the manuscript and the time spent on reviewing our article. We took all the comments very seriously and we believe that they contributed to the improvement of the quality of our work.

Below we present the answer to the review.

Point 1: - Although the last paragraph of the introduction part should introduce the aim of your review article and explain the importance of this study, I could not find the proper description there. Please talk about why this topic was chosen as a narrative review and describe what you discussed.

Response 1: We expanded on this topic in the introduction.

Point 2: In my opinion, the order of the manuscript is confusing and repeated. I advise you to change this order according to the following: first, define each type of GI, then define the role of ADAM in GIs, after that, it is better to define the role of miRs in GIs, and finally talk about ADAM/related miRs in GIs.

Response 2: Thank you very much for this attention. We also took into account the order of descriptions, however, the other two reviewers of our work suggested the order of descriptions as it is now, which is why we decided to leave this form.

Point 3: The conclusion part does not include references or any table. I advise you to add a new session and separate a new conclusion part.

Response 3: The conclusion includes table 3, which applies the role of ADAM/MiR axis in GI.

Point 4: The references were manually entered and their format is not similar. Please complete reference 9. I also kindly suggest you add references via Endnote/Mendeley that could prevent multiple errors.

Response 4: We corrected mentioned points.

Point 5: This article includes 3 tables but lacks proper descriptions and the definition of abbreviations. In the conclusion part, you mentioned table 2 instead of table 3. Please correct it.

Response 5: We added the definition of abbreviations and corrected mistake of mentioned table.

Round 2

Reviewer 2 Report

Comments and Suggestions for Authors

Author should provide a graphical presentation of the role of ADAMs with miRNA in the development of different GIT cancers (1 or 2 figures).

Author Response

Response to Reviewer 2 Comments

Thank you very much for any comments on the manuscript and the time spent on reviewing our article. Below we present the answer to the review.

Author should provide a graphical presentation of the role of ADAMs with miRNA in the development of different GIT cancers (1 or 2 figures).

The comment is valid, but the authors are concerned that the paper is becoming too large and the inclusion of a figure may not be necessary. Please respect our opinion.

Round 3

Reviewer 2 Report

Comments and Suggestions for Authors

Accept